# How can new energy development reduce $CO_2$ emissions: Empirical evidence of inverted U-shaped relationship in China

**Feng Xiong** [1]*, **HuiDong Mo** [2]

**1** School of Economics and Management, Chongqing University of Posts and Telecommunications, Chongqing, China, **2** School of Economics and Management, Heilongjiang University, Harbin, China

* xiongfeng@cqupt.edu.cn

**Data Availability Statement:** The data set of this paper are upload in Harvard Dataverse, and the DOIs is https://doi.org/10.7910/DVN/YDX6X9.

**Funding:** This study was supported by Humanities and Social Science Research of Chongqing Education Commission (No: 22SKGH129) and

## Abstract

This article is based on the statistical yearbook data of 30 provinces, municipalities and autonomous regions in China (excluding Hong Kong, Macao, Taiwan, and Tibet Autonomous Region) from 2000 to 2017, a total of 18 years of statistical yearbook data was used to conduct in-depth research on the reduction of $CO_2$ emissions from the development of new energy in the region. First, it is proposed that the regional new energy development has a significant negative effect on $CO_2$ emissions. Meanwhile, this impact has a significant time lag effect, and the development of new energy cannot be quickly and effectively applied in the short term to replace traditional fossil energy in the dynamic model. Therefore, there is a significant positive impact in the short term, but the significant negative effect of new energy development on $CO_2$ emission can be shown in the long run. Secondly, the new energy development has a significant non-linear impact on $CO_2$ emissions, showing an inverted U-shaped relationship, which confirms the existence of the Environmental Kuznets Curve (EKC) of $CO_2$ emissions based on new energy development. Finally, in order to alleviate the continuous impact of national economic development on $CO_2$ emissions, the DID model is used to prove that the level of technological innovation has a significant moderating effect on the $CO_2$ emission reduction effect of new energy development, which confirms theoretically the importance of technological innovation in accelerating new energy substitution and improving energy efficiency.

## 1. Introduction

Since China's reform and opening up in the late 1970s, China's economy has taken off and energy consumption has grown rapidly. Q. Zhang et al. [1] Based on panel data of 30 provinces in China, the study found that per capita GDP and per capita energy consumption have a significant positive relationship, and the secondary industry plays a role in the growth of residential energy consumption important role. At the same time, with the continuous growth of China's economy and the continuous increase of energy consumption, $CO_2$ emissions will further increase, especially in the case of increasingly severe global warming, the future will face

China Postdoctoral Science Foundation (No: 2023M730388). The funders had no role in study design, data collection and analysis, decision to publish, or preparation of the manuscript.

**Competing interests:** The authors have declared that no competing interests exist.

huge pressure to reduce emissions. Therefore, in the face of the climate crisis brought about by global warming and the international community's emphasis on low-carbon energy development, China has accelerated the pace of building an environment-friendly society. To this end, China has also set a goal of increasing the proportion of non-fossil energy in primary energy consumption from 15% in 2015 to 20% in 2030 and committed to the international community that the carbon generated by GDP in 2030 will be higher than that in 2005. 60% lower. This makes the reduction of $CO_2$ emissions gradually become a hot issue in academic research in the environmental field, and also the core of the government's environmental protection strategy work. There are many factors or measures to reduce $CO_2$ emissions in the current literature research, such as green supply chain networks [2], policy regulations and technological innovation [3]. Hu et al. [4] found that, while maintaining steady economic development, energy intensity and energy structure are the key factors driving the decoupling of GDP and $CO_2$ emissions. At the same time, new energy has a significant crowding-out effect on traditional fossil energy, so the optimization of the energy structure will help $CO_2$ emissions get rid of the direct impact of economic development. Based on China's data from 2004 to 2017, [5] uses the autoregressive distributed lag (ARDL) model to confirm that personal new energy consumption also has a significant positive effect on economic growth. Therefore, the development of new energy can replace traditional fossil energy to a certain extent, and it may gradually become a key factor for China to ensure steady economic growth and achieve $CO_2$ emission reduction targets as scheduled.

So, can the development of new energy effectively reduce the $CO_2$ emissions of Chinese provinces and achieve green and rapid economic growth in China? In response to this problem, this paper is based on China's provincial panel data, combined with the panel data model to deeply study the linear, nonlinear and dynamic relationship between the development of new energy and $CO_2$ emissions in China's provinces, and uses the DID model to reveal the impact of technological innovation on new energy. The regulatory effect of energy development in reducing $CO_2$ emissions. Therefore, this paper not only expounds on the basic situation of my country's new energy development since 2000 from the perspective of data, but also empirically tests the impact mechanism of new energy development on $CO_2$ emissions, to promote the rational development of new energy and achieve carbon neutrality and carbon peaking in my country Objectives provide a solid basis.

In summary, this paper adds to the existing researches on the relationship between new energy development and carbon emissions from the following three main aspects: 1) The new energy development in China has already realized a $CO_2$ emission reduction effect. 2) The impact of new energy development on carbon emissions is non-linear, i.e. Environmental Kuznets Curve (EKC) of $CO_2$ emissions based on new energy development exists. 3) Technological innovation can contribute to the $CO_2$ emissions reduction effect of the new energy development.

## 2. Literature review

At present, Y.-J. Zhang & Da [6] pointed out that economic growth significantly promotes the growth of $CO_2$ emissions, the cleanliness and reduction of energy structure and energy intensity are two key elements in promoting the decoupling of $CO_2$ emissions and economic development, in addition, the second industry accounts for the largest $CO_2$ emissions, which has great potential for emission reduction. Feng et al. [7] believed that energy structure and energy intensity are the key factors affecting $CO_2$ emissions, and the improvement of energy intensity can promote the growth of traditional energy utilization efficiency. From the thermodynamic balance theory, new energy cannot completely replace fossil energy, which also conforms to

the core of new energy development that optimizes the energy structure of fossil energy mainly to improve energy efficiency, promote extrusion of fossil energy consumption, make the pollution of energy consumption generated caused by human activity within a reasonable range to achieve the purpose of reducing $CO_2$ emissions and the mutually beneficial and effective development of economic development and environmental optimization. Ren et al. [8] also found that economic growth plays a vital role in promoting $CO_2$ emissions. In this case, a decrease in energy intensity can significantly inhibit a rise in $CO_2$ emissions, and clean energy structure is conducive to improving the proportion of new energy in the total energy volume. Therefore, new energy which can integrate economic growth and sustainable energy utilization not only maintains the energy consumption of economic development, but also promotes the sustainability of energy.

Gong et al. [9] found the biggest driver of $CO_2$ emissions growth is the growth rate of GDP per capita, and $CO_2$ emissions in China are closely related to economic development and resident living standards. Economic development is accompanied by $CO_2$ emissions increased, but the growth is not entirely dependent on $CO_2$ emissions. Fakhri et al. [10] also consider the pollution-free sectors with economic growth potential rather than traditional energy consumption sectors. Taking G20 countries as research objects, Pao & Chen [11] found the possibility of absolute decoupling between environmental pressure and economic growth in the case of declining environmental pressure and sustained economic growth, while identifying new energy as the only way to achieve absolute decoupling. Obviously, the development of new energy is an important way to reduce $CO_2$ emissions. Li et al. [12] took OCED countries as the research object and found that it is crucial to make a national energy development strategy by promoting new energy development and research in a large number of traditional energy countries. Current global economic growth is in a sub-optimal state, the direction of technological innovation is gray-oriented, and there is always too little effort to invest in green research [13]. Green technology innovation is one of the most effective ways to achieve the goal of net zero emissions [14]. The new energy sector, as an important component of green technology, has revolutionized the production and use of energy [15].

As for the relationship between $CO_2$ emissions and the economy, American economists G. Grossman and A. Kureger believed the inverted U between economic growth and environmental pollution, namely the evolution of environmental quality with the accumulation of economic growth, which is known as the EKC curve. A large number of scholars have verified and explored whether the EKC curve is consistent with the current actual conditions of China. Jalil & Mahmud [16] proposed that China's economic growth shows a one-way causal relationship with $CO_2$ emissions. Meanwhile, China's economic growth has a positive linear effect on $CO_2$ emissions, while the nonlinear effect is an inverted U-shape [17]. Sun et al. [17] have found that China 's EKC curve is "N-shape " rather than "inverted U-shape". It indicated that China's green economic development is still in the transition period and cannot reach the stage of coordinated economic and environmental development. Some studies still point out that China's Kuznets curve does not conform to the inverted U-shape assumption of the EKC curve, mainly because environmental protection has not grown along with China's economic development [18].

Rehman et al. [19] researched Pakistan's new energy and found that renewable energy generation can promote long-term GDP growth per capita. The government should strengthen the use of renewable energy (wind, water, nuclear power, and so on) to address the country's energy crisis. Undoubtedly, the development of new energy is an urgent need for worldwide economic development today. For China, can the new energy development in various provinces effectively reduce $CO_2$ emissions? In recent years, there has been some literature pointing out that the promotion effect of new energy on economic growth was more significant

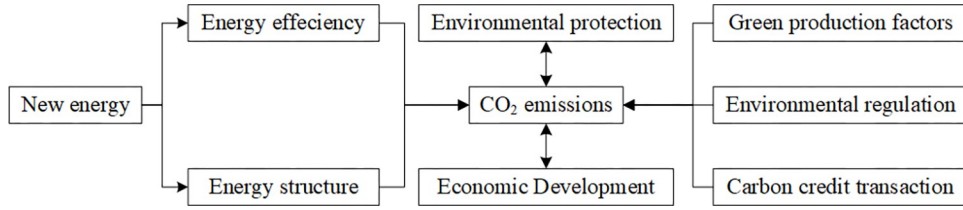

**Fig 1. The relationship between new energy development and CO₂ emissions.**

than that of traditional fossil energy, and new energy consumption in Sichuan Province provided a greater boost (at about 230%) to economic growth compared with its traditional mineral energy consumption. Economic activity is a key element of $CO_2$ emissions growth in Jiangsu Province, and the energy intensity effect plays a core role in the reduction of $CO_2$ emissions [20]. China's less-developed provinces have caused relatively higher $CO_2$ emissions due to the low level of clean technology, while the eastern developed regions can reduce the lower intensity of $CO_2$ emissions because there is economic strength to improving energy use efficiency [21].

Since the reform and opening up, China has dominated the extensive economic growth and promoted great economic development at the expense of the environment and resources. However, behind the prosperity, the environmental quality has dropped sharply, and the environmental problems directly threaten economic growth, thus inhibiting economic development to a certain extent. Since General Secretary Xi underscored the importance of ecological conservation and put forward the concept that lucid waters and lush mountains are invaluable assets, the authorities have issued relevant policies to support the development of green energy and other environment-friendly industries. As shown in Fig 1, the development of new energy not only improves energy efficiency [22], but also changes the national energy structure [23], which greatly inhibits the growth of $CO_2$ emissions. There are other related studies that explore the factors that reduce $CO_2$ emissions from the perspective of environmental protection and high quality economic development: 1) the action mechanism of green total factor productivity and $CO_2$ emissions [24–26]; 2) the action mechanism of environmental regulation and $CO_2$ emissions [27, 28]; 3) the regulatory control of carbon credit transaction [29, 30].

To sum up, the academic conclusion on new energy, $CO_2$ emissions and economic development has not yet been formed. In the development stage of new energy, there is still a large space for researches on $CO_2$ emissions and economic development. Therefore, this paper not only aims to confirm whether the linear relationship between new energy development and $CO_2$ emissions is positive or negative under China's provincial panel data. It also has to explore whether there is a significant nonlinear relationship between new energy development and $CO_2$ emissions. That is, whether the Kuznets curve of environmental pollution based on new energy development exists. Meanwhile, the authors examine the role of other factors (e.g., technological innovation) in moderating the impact of new energy development on reducing $CO_2$ emissions.

## 3. Research hypothesis and model design

### 3.1 Hypothesis

Boqiang et al. [31] suggest that the development of nuclear, hydropower, wind power and other new energy sources can effectively reduce the pressure on fossil energy and reduce $CO_2$ emissions, while carbon capture and carbon storage technologies are also viable options.

However, due to technical and productivity constraints, carbon capture and carbon storage technologies are risky and costly, and can accelerate energy shortages and hinder the sustainable development of national economies [32]. Therefore, the development and use of new energy sources plays a crucial role in reducing CO$_2$ emissions. Therefore, the first hypothesis of this paper is proposed.

**H1: Ceteris paribus, the new energy development has a significant negative effect on CO$_2$ emissions based on China's provincial panel data.**

The rapid development of related industries driven by the development of new energy is bound to increase energy consumption and reflect the economic scale effect, which can improve CO$_2$ emissions [33]. However, as the level of new energy development continues to improve, its effect on the reduction of CO$_2$ emissions will gradually increase. So, is there an EKC hypothesis between new energy development and CO$_2$ emissions in China? Sinha & Shahbaz [34] attempted to estimate the EKC for CO$_2$ emission in India for the period of 1971–2015. The study found that the renewable energy has a significant negative impact on CO$_2$ emissions, whereas for overall energy consumption, the long run elasticity is found to be higher than short run elasticity. Therefore, the second hypothesis in this paper has been provided.

**H2: Ceteris paribus, the negative impact of the new energy development on CO$_2$ emissions is inverted U-shape, i.e. the impact gradually decreases as the new energy development increasing.**

Energy technology is the breakthrough of the new round of technological and industrial revolution [35]. As people's awareness of environmental protection increases and energy consumption accelerates worldwide, the new round of energy revolution will shift from the scale efficiency of industrial civilization to the green and low-carbon energy era in which the scale is set by efficiency in the information age. Therefore, technological innovation plays a decisive role in the energy revolution [36], and it is an inexhaustible driving force for the optimization of the energy structure and its transformation and upgrading [37]. Only by mastering core technologies through innovation and building a modern energy system that is clean, low-carbon, safe and efficient can we seize the key to energy change and grasp the initiative for sustainable and healthy energy development. So, does the level of regional technological innovation have a significant moderating effect on the CO$_2$ emission reduction effect of new energy development? Based on Chinese provincial panel data from 2006–2015, Pan et al. [38] used directed acyclic graph (DAG) and structural vector autoregressive (SVAR) models and found that technological innovation plays an important role in promoting energy efficiency in the short and long term. Therefore, the third hypothesis of this paper is proposed.

**H3: Ceteris paribus, the negative effect of the new energy development on CO$_2$ emissions is more significant in provinces with higher level of technology innovation.**

### 3.2 Empirical model

**3.2.1 Multiple regression model.** In order to analyze the mechanism of the impact of new energy development on CO$_2$ emissions, a simple linear mathematical model between multiple variables was first developed, as in Eq (1). The statistical analysis of Eq (1) was carried out using panel data to obtain the estimates and significance of the parameters and test the strength and significance of the influence of the explanatory variables on CO$_2$ emissions in each province.

$$Y = \beta_0 + \beta_1 X_1 + \beta_2 X_2 + \cdots + \beta_p X_p + \varepsilon \qquad (1)$$

where $\beta_0,\beta_1,...,\beta_p$ denote the regression coefficients. $\varepsilon$ denotes the error term, which follows a normal distribution with a mean of 0 and a standard deviation of $\delta$. In this paper, the maximum likelihood (MLE) method will be used to statistically analyse Eq (1) and obtain estimates of the regression coefficients. Multiple regression analysis ignores time (annual) trends, and the introduction of the estimation method of feasible generalized least squares (FGLS) can not only explain the time trends of the data to a certain extent, but also solve the heteroskedasticity problem of the multiple regression model. Therefore, the FGLS method significantly outperforms the mixed regression results in the multiple regression model for face-to-face panel data, as evidenced by the F-test.

**3.2.2 Fixed effects models.** Panel data is a sample of observations of different study individuals over time trends, with both time series and cross-sectional dimensions. Therefore, in order to fit the existence of time effects and individual effects between variables in the data, an effect model is introduced, the mathematical model of which is as follows.

$$Y_{it} = \alpha + \beta X_{it} + \delta control_{it} + c_i + p_t + \varepsilon_{it} \tag{2}$$

$i$ denotes the country in the sample and $t$ denotes the period of the data from 2000 to 2017. $Y_{it}$ denotes the explanatory variable, i.e. $CO_2$ emissions of province $i$ in year $t$. $X_{it}$ is new energy generation, and $control_{it}$ denotes the other control variables. $c_i$ represents the fixed trend of each province, which is named the fixed effect. $p_t$ represents the time trend common to all provinces, which is named the time effect. $\varepsilon_{it}$ is the error term. $\beta$ and $\delta$ are both parameters of the regression estimates. When $c_i$ is correlated with $X_{it}$ and $control_{it}$, it is a fixed utility model. While, $c_i$ is randomly distributed and follows some normal distribution, and it is a random utility model.

In statistics, there are various statistical estimation methods to calculate the regression coefficients of the explanatory variables. In order to make Eq (2) have a better fit, the main model is regressed using least squares dummy variable (LSDV) estimation when only fixed effects (provincial trend) are considered, while maximum likelihood estimation (MLE) is used when both fixed effects and time utility are combined. Finally, this paper not only compares the goodness of fit of the traditional multiple regression model and the fixed utility model through the F-test, but also uses the Hausman test to examine the differences between the fixed utility model and the random utility model.

# 4. Empirical analysis

## 4.1 Data resource

Based on the database from the China Statistical Yearbook and the China Energy Statistics Bureau, this paper selects the annual data from 2000 to 2017 for 30 provinces, cities and autonomous regions in China (excluding Hong Kong, Macao, Taiwan and Tibet Autonomous Region, hereafter collectively referred to as provinces). The annual $CO_2$ emissions of each province were measured by the Intergovernmental Panel on Climate Change (IPCC), where $CO_2$ is mainly generated from fossil energy combustion.

## 4.2 Variables definition

**4.2.1 The measurements of $CO_2$ emissions.** In this paper, $CO_2$ emissions are estimated based on the amount of energy fuels and combustion emission factors. The data on the various energy sources consumed in the 30 provinces from 2000 to 2017 were obtained from the China Energy Statistical Yearbook from 2000 to 2017, and the (net) consumption of the 30 provinces from 2000 to 2017 was obtained by excluding the inputs and losses in the process of energy conversion and the part used as raw materials in industrial production.

This was estimated using Eq (3).

$$CO_2 = \sum_{i=1}^{14} CO_{2i} = \sum_{i=1}^{14} E_i \cdot NCV_i \cdot CEF_i \tag{3}$$

Where $CO_2$ represents the $CO_2$ emissions to be estimated. $i$ represents various energy fuels, and $E_i$ represents the combustion consumption of various energy sources. $NCV_i$ is the average low-level heat of energy sources, which is used to convert various energy consumption into energy units (TJ). $CEF_i$ represents the $CO_2$ emission factor of various energy sources, which is calculated as in Eq (4).

$$CEF_i = CC_i \cdot COF_i \cdot \left(^{44}/_{12}\right) \tag{4}$$

**4.2.2 The measurements of new energy development.** At present, the main source of new energy is concentrated in power generation, and thermal power is a traditional energy source. Therefore, the new energy production of each province in this paper is calculated by the following formula:

$$NEG_{it} = EG_{it} - TEG_{it} \tag{5}$$

In the above equation, $EG_{it}$ denotes the total electricity generation in province $i$ in year $t$, and $TEG_{it}$ denotes the total thermal power generation. Therefore, $NEG_{it}$ denotes the new energy development in province $i$ in year $t$, which includes the total production of new energy such as water, wind, nuclear and photovoltaic power generation.

## 4.3 Statistical description of data

Table 1 describes the statistical results of the data, which contains panel data for a total of 18 years from 2000–2017 for 30 provinces in China. The new energy development, which is measured by the share of new energy generation, is the explanatory variable with a range of (0.0000, 0.9189) and a mean value of 0.2322, while the other control variables cover the areas of economic, social, environmental, demographic, and so on. Technology innovation is

**Table 1. Statistical description of data.**

| Variable type | Definition | Obs. | Mean | Std. | Min | Max |
|---|---|---|---|---|---|---|
| Explained variable | The log of $CO_2$ emissions | 540 | 9.8219 | 0.8628 | 6.7074 | 11.4606 |
| Explanatory variable | New energy development | 522 | 0.2322 | 0.2306 | 0 | 0.9189 |
| Control variable | The log of total electricity generation | 522 | 6.7583 | 0.8967 | 3.6648 | 8.5809 |
| | The log of forest area | 420 | 5.9150 | 1.4257 | 0.6366 | 7.8192 |
| | The log of population | 540 | 8.1555 | 0.7571 | 6.2480 | 9.3209 |
| | The log of road mileage | 540 | 11.2699 | 0.8999 | 8.6945 | 12.7069 |
| | The log of land area | 540 | 11.9279 | 1.2304 | 6.8565 | 14.3253 |
| | Unemployment rate | 539 | 3.5596 | 0.7112 | 0.8 | 6.5 |
| | Consumer price index | 540 | 102.2639 | 2.0347 | 96.7 | 110.1 |
| | GDP growth rate | 540 | 10.9275 | 2.8306 | -2.5 | 23.8 |
| | Telephone penetration | 469 | 62.9576 | 36.5361 | 2.12 | 189.5 |
| | Internet penetration | 483 | 11.7180 | 12.8503 | 0.067 | 80.24 |
| | Share of secondary industry | 540 | 45.6201 | 7.8729 | 19 | 59.3 |
| Moderating variable | Technology innovation | 540 | 8.7320 | 1.6778 | 4.2485 | 12.7149 |
| Instrumental variables | The natural logarithm of pm2.5 | 540 | 39.2957 | 15.5335 | 7.1000 | 84.3000 |
| | The natural logarithm of thermal power generation | 522 | 6.4249 | 0.9769 | 3.2619 | 8.5454 |

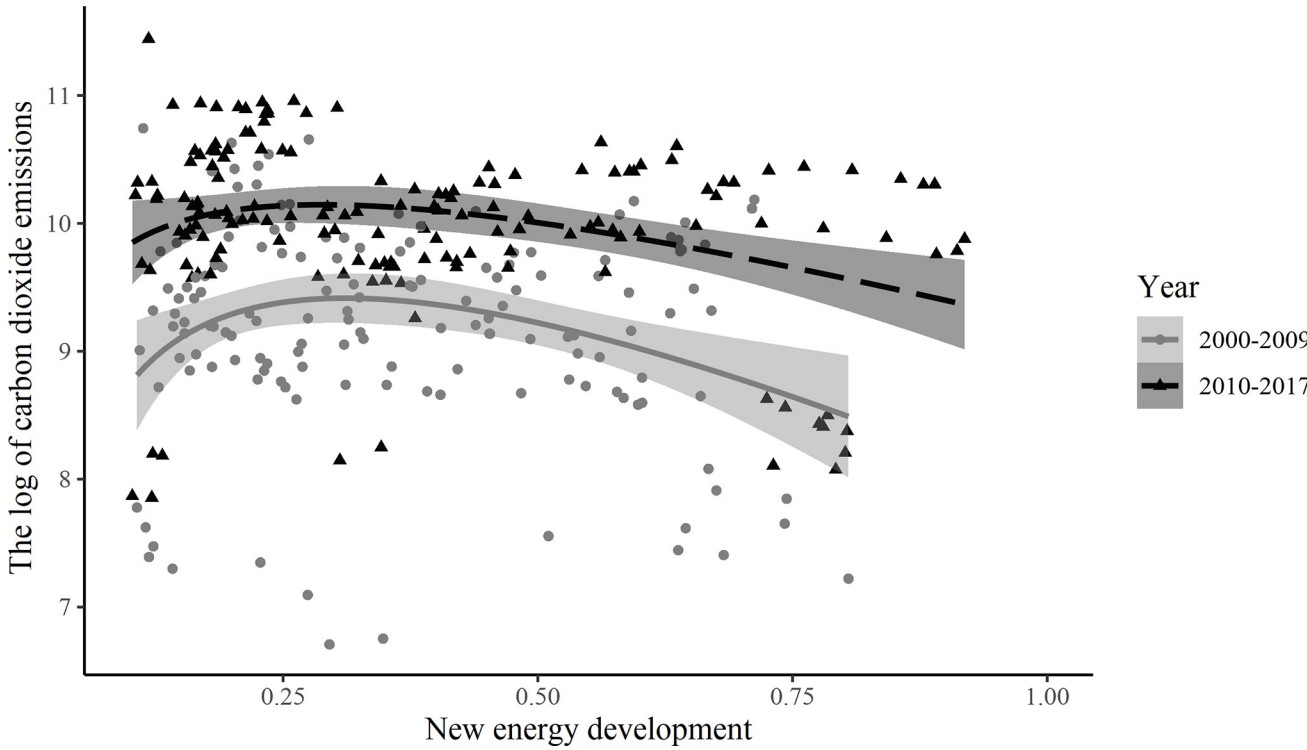

**Fig 2. Scatter distribution and fitting curve of the new energy development and the natural logarithm of CO$_2$ emissions.**

moderating variable, which is measured by the natural logarithm of the number of patent applications. And the natural logarithm of pm2.5 and thermal power generation, which are highly correlated with new energy development in the correlation test, are used as instrumental variables to test endogeneity of the original models.

Fig 2 depicts the scatter distribution and fitted curves between the new energy development and CO$_2$ emissions. The scatter distribution shows that CO$_2$ emissions are significantly lower in the 2000–2009 time period than in the 2010–2017 time period, i.e. the development of the national economy as a whole has increased CO$_2$ emissions. At the same time, the trend of both fitted curves is accelerating downwards, explaining that the new energy development can reduce CO$_2$ emissions, and that this effect is gradually increasing with new energy development improving. Comparing the two curves, the curvature of the curve is greater in the 2000–2009 time period than in the 2010–2017 time period, which means that the effect of new energy on CO$_2$ emissions reduction is relatively stronger in the 2000–2009 time period. Thus, a simple description of the data shows that the relationship between new energy development and CO$_2$ emissions in China is consistent with the inverted U-shaped hypothesis of the EKC curve.

### 4.4 Statistical test

Based on data, this paper researches the impact mechanism of new energy development on CO$_2$ emissions in each province. First, preliminary statistics such as unit root and multicollinearity are performed. As for the unit root test, to find potential problems. This paper adopts the method of Fisher's test. Fisher's test does not require strongly balanced panel data and uses four different transformations of inverse chi-square, inverse normal, inverse logarithmic transformation and modified inverse chi-square through aggregate statistics and P-values. To

**Table 2. Unit test.**

| | | New energy development | | The log of $CO_2$ emissions | |
|---|---|---|---|---|---|
| | | Statistic | P-value | Statistic | P-value |
| Inverse chi-squared (240) | P | 121.3046 | 0.0000 | 118.6697 | 0.0000 |
| Inverse normal | Z | -1.7873 | 0.0369 | -3.6589 | 0.0001 |
| Inverse logit t (604) | L* | -2.0867 | 0.0193 | -4.1896 | 0.0000 |
| Modified inv. chi-squared | Pm | 5.8777 | 0.0000 | 5.3558 | 0.0000 |

perform a unit root test on the main explanatory variable and the explained variable [39]. The results in Table 2 show that there is no unit root problem for the two data variables of new energy development and the log of $CO_2$ emissions in each province, that is, the data are stable in the time series.

Multicollinearity, or strong correlations between variables, can distort or inaccurate model estimates. Therefore, we performed a test for multicollinearity, a maximum inflation factor (VIF) proposed by [40]. VIF is the diagonal element of the inverse of the correlation matrix (standard matrix). Practical experience has shown that if any of the VIFs exceeds 5 or 10, it indicates that the associated regression coefficients are underestimated due to multicollinearity [41]. Table 3 shows that there is no multicollinearity among the explanatory variables, and the constructed model is valid and consistent.

## 4.5 Regression analysis of the impact of new energy development on $CO_2$ emissions

**4.5.1 Static linear relationship.** Using the econometric model in Section 3.2 and the panel data for regression estimation, the results in Table 4 were obtained. Model (1), a simple multiple regression, first explains that new energy development has a significant negative effect on $CO_2$ emissions and that a one percentage point increase in new energy development is associated with a 0.9946% reduction in $CO_2$ emissions. The results of model (2), which introduces a feasible generalized least squares approach to account for heteroskedasticity and time trends, still show that new energy development has a significant negative effect on $CO_2$ emissions, and the magnitude of the effect is very similar to that of model (1). Since the panel data have strong individual trends, models (3)-(5) introduce effect models to control for fixed trends in each province in each year, and obtain results for fixed effects, random effects and fixed effects

**Table 3. Multicollinearity test.**

| Variables | VIF | SQRT VIF | Tolerance | R-Squared |
|---|---|---|---|---|
| New energy development | 1.71 | 1.31 | 0.5851 | 0.4149 |
| The log of total electricity generation | 6.64 | 2.58 | 0.1507 | 0.8493 |
| The log of forest area | 5.32 | 2.31 | 0.1880 | 0.812 |
| The log of population | 5.84 | 2.42 | 0.1712 | 0.8288 |
| The log of road mileage | 5.06 | 2.25 | 0.1978 | 0.8022 |
| The log of land area | 4.36 | 2.09 | 0.2293 | 0.7707 |
| Unemployment rate | 1.64 | 1.28 | 0.6083 | 0.3917 |
| Consumer price index | 1.12 | 1.06 | 0.8916 | 0.1084 |
| Growth rate of GDP | 2.10 | 1.45 | 0.4764 | 0.5236 |
| Telephone penetration | 4.31 | 2.08 | 0.2320 | 0.768 |
| Internet penetration | 4.06 | 2.01 | 0.2464 | 0.7536 |
| Share of secondary industry | 2.36 | 1.53 | 0.4246 | 0.5754 |
| Mean VIF | 3.71 | | | |

**Table 4. The impact of new energy developments on $CO_2$ emissions.**

| | (1) | (2) | (3) | (4) | (5) |
|---|---|---|---|---|---|
| | OLS | FGLS | FE | RE | FE |
| New energy development | -0.9946*** | -0.7785*** | -0.7670*** | -0.8882*** | -0.7031*** |
| | (-17.14) | (-12.18) | (-6.77) | (-9.00) | (-6.29) |
| The log of total electricity generation | 0.5976*** | 0.5427*** | 0.6124*** | 0.6298*** | 0.4601*** |
| | (24.46) | (20.42) | (22.53) | (25.12) | (13.30) |
| The log of forest area | -0.0798*** | -0.0690*** | 0.1114*** | -0.0145 | 0.0009 |
| | (-4.52) | (-4.74) | (2.95) | (-0.56) | (0.02) |
| The log of population | 0.3316*** | 0.3705*** | -0.3531** | 0.1947*** | -0.4164*** |
| | (11.75) | (12.34) | (-2.48) | (3.63) | (-3.00) |
| The log of road mileage | 0.1241*** | 0.0520** | 0.1536*** | 0.1495*** | 0.0375 |
| | (4.45) | (2.47) | (6.10) | (5.96) | (1.11) |
| The log of land area | 0.0332* | 0.0325** | 0.0250 | 0.0068 | 0.0329* |
| | (1.88) | (2.33) | (1.38) | (0.39) | (1.96) |
| Unemployment rate | 0.0816*** | 0.0533*** | -0.0012 | -0.0004 | 0.0116 |
| | (4.77) | (3.17) | (-0.06) | (-0.02) | (0.64) |
| Consumer price index | -0.0053 | -0.0121** | 0.0019 | -0.0015 | -0.0275*** |
| | (-0.85) | (-2.04) | (0.62) | (-0.47) | (-3.40) |
| Growth rate of GDP | -0.0114** | 0.0019 | -0.0016 | 0.0015 | 0.0017 |
| | (-2.53) | (0.54) | (-0.60) | (0.57) | (0.45) |
| Constant | 2.3603*** | 3.5988*** | 5.8966*** | 2.6297*** | 12.0366*** |
| | (3.64) | (5.45) | (5.36) | (5.10) | (8.92) |
| Observation | 406 | 406 | 406 | 406 | 406 |
| R-squared | 0.926 | | 0.864 | 0.8556 | 0.891 |
| Province effect | No | No | Yes | Yes | Yes |
| Year effect | No | Yes | No | No | Yes |
| Hausman test | | | 24.54*** | – | |

Note

\*, \*\* and \*\*\* denote 10%, 5% and 1% respectively, and t-statistics for the corresponding explanatory variables are shown in parentheses. (Same as below)

while controlling for time trends, respectively. The significant results of the Hausman test indicate that the fixed effects model will be due to the random effects model, so the analysis in the later section will be dominated by the fixed effects model in models (3)-(5), where new energy development has a consistent negative direction on $CO_2$ emissions, and **H1** is confirmed.

**4.5.2 Dynamic effects and non-linear relationships.** The above analysis has demonstrated that new energy development in China's provinces has a significant negative effect on $CO_2$ emissions, but does this effect have a dynamic moderating effect? Therefore, this section adds the dynamic lagged terms of the explanatory variables to the static model to obtain the dynamic panel model, which is shown in Eq (6). The model not only explains the time trends ($p_t$) and individual trends ($c_i$), but also accounts for the fact that the explanatory variables are dynamically influenced by their previous period. At the same time, a portion of the influences omitted in the current period is included in the dynamic lagged term of the explained variable, so the dynamic model can mitigate to some extent the endogeneity problem arising from omitted variables and increase the consistency and explanatory power of the model results.

$$Y_{it} = \gamma Y_{it-1} + \alpha + \beta X_{it} + \delta control_{it} + c_i + p_t + \varepsilon_{it} \tag{6}$$

**Table 5. Dynamic effects and non-linear effects of new energy development on $CO_2$ emissions.**

|  | (1) | (2) | (3) | (4) | (5) | (6) |
|---|---|---|---|---|---|---|
| First order lag term of log of $CO_2$ emissions | 0.5117*** | 0.5287*** |  |  |  | 0.4364*** |
|  | (12.10) | (12.28) |  |  |  | (10.08) |
| New energy development | -0.3912*** |  | -0.5604*** | -0.8985*** | 0.7413*** | 0.4383** |
|  | (-4.01) |  | (-4.14) | (-7.00) | (3.57) | (2.36) |
| First order lag term of new energy development |  | -0.2298** | -0.2487* |  |  |  |
|  |  | (-2.34) | (-1.86) |  |  |  |
| First order differential term of new energy development |  |  |  | 0.2921** |  |  |
|  |  |  |  | (2.14) |  |  |
| The squared of new energy development |  |  |  |  | -1.4614*** | -0.8857*** |
|  |  |  |  |  | (-8.02) | (-5.19) |
| control variables | ✓ | ✓ | ✓ | ✓ | ✓ | ✓ |
| Constant | 5.7172*** | 5.3257*** | 12.1010*** | 5.9997*** | 11.6885*** | 6.4359*** |
|  | (4.57) | (4.20) | (9.00) | (5.47) | (9.40) | (5.30) |
| Observation | 406 | 406 | 406 | 406 | 406 | 406 |
| R-squared | 0.923 | 0.921 | 0.892 | 0.866 | 0.908 | 0.928 |
| Province effect | Yes | Yes | Yes | Yes | Yes | Yes |
| Year effect | Yes | Yes | Yes | Yes | Yes | Yes |

When the first-order difference term of the explanatory variable ($X_{it}$) is introduced based on the static model, Eq (7) is obtained, where the regression coefficient $\lambda$ of $\Delta X_{it}$ represents the short-term impact of new energy on $CO_2$ emissions, while the regression coefficient $\beta$ of $X_{it}$ represents the long-term impact.

$$Y_{it} = \lambda \Delta X_{it} + \alpha + \beta X_{it} + \delta control_{it} + c_i + p_t + \varepsilon_{it} \tag{7}$$

The regression results in Table 5 show that the first-order lag term of $CO_2$ emissions has a significant positive effect on the current period, i.e. $CO_2$ emissions are affected by their lags in time. Meanwhile, models (2) and (3) demonstrate that the first-order lagged term of new energy development has a significant negative effect on $CO_2$ emissions under the dynamic model and static model respectively, i.e. there is a significant lag effect. In model (4), the first-order difference term for the share of new energy generation (short-term effect) is significantly positive, while new energy development (long-term effect) is significantly positive. This indicates that the new energy development cannot be applied quickly enough to replace traditional fossil energy sources in the short term, so it shows a significant positive effect in the short term, but in the long term the effect of new energy development on $CO_2$ reduction is significant.

The negatively significant regression results of introducing the squared term of new energy development into the static and dynamic models in models (5) and (6) respectively indicate that new energy development has a significant non-linear effect on $CO_2$ emissions and shows an inverted U-shaped relationship in the model (as shown in Fig 3). Then, **H2** was verified. The new energy industry is a highly capital-invested and technology-intensive industry, and at a relatively low level, the investment in its industry raises the economic development of the region, which in turn contributes to $CO_2$ emissions to a small extent, i.e. new energy does not fundamentally change the energy mix, and fossil energy remains dominant. Therefore, the effect of new energy development in reducing $CO_2$ by replacing fossil energy is lower than the positive effect it brings to $CO_2$ by promoting economic development, which makes the overall effect of new energy development not to reduce $CO_2$ emissions. When the new energy

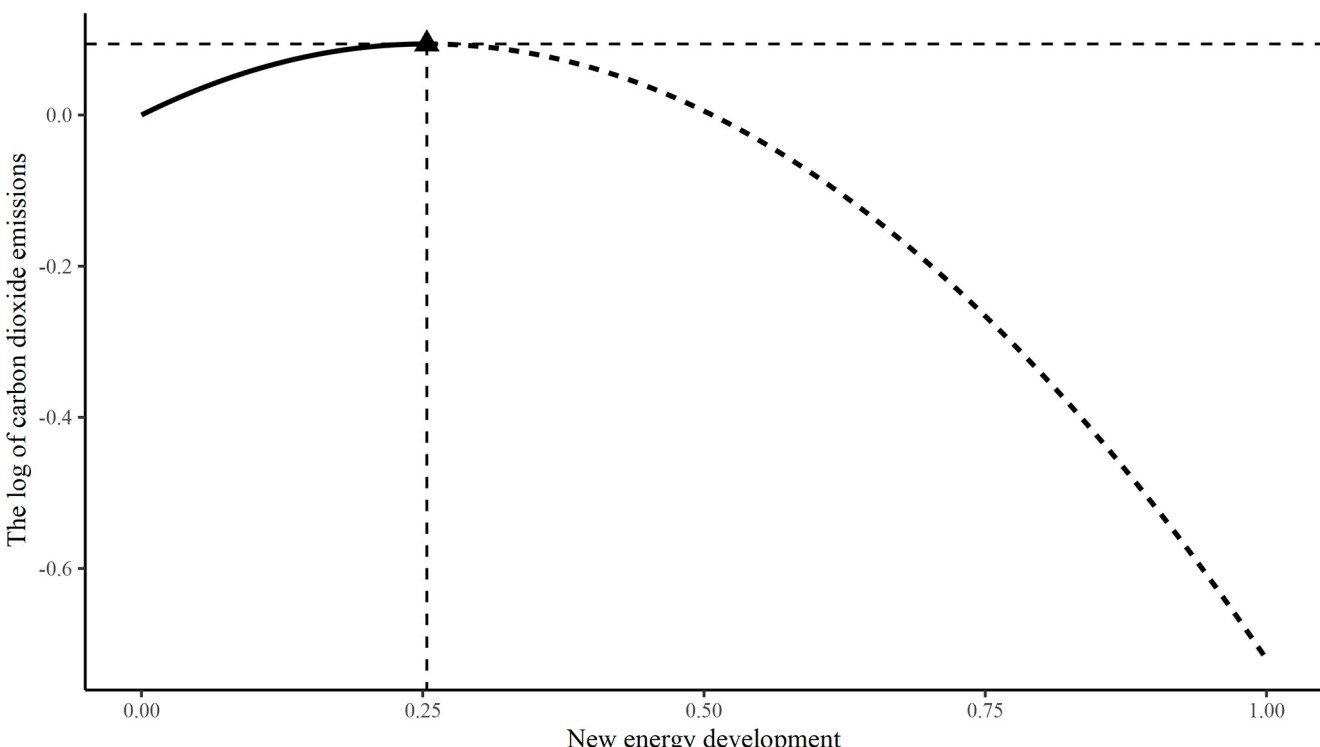

**Fig 3. The non-linear effect of new energy development on $CO_2$ emissions.**

development reaches a certain stage (25%), the new energy gradually takes the main part in the energy development to a low position, then the negative effect of the new energy development on $CO_2$ emissions will increase with the new energy development, that is, after the new energy development and utilization reaches a certain level, the effect of the new energy development to reduce $CO_2$ emissions is a marginal effect increasing.

**4.5.3 The moderating role of the level of technological innovation.** In this paper, the dummy variables based on the grouping of technological innovation levels were used to divide the sample into high technology innovation regions (>8.7320) and low technology innovation regions (≤8.7320), using the mean value of technology innovation data (8.7320) as the threshold, and then comparing and analyzing the difference in the impact of the share of new energy generation on $CO_2$ emissions under different technology innovation levels. In model (1) of Table 6 for the low technology innovation region (≤8.7320), the effect of the share of new energy generation on $CO_2$ emissions is negative and significant, with a value of -0.2441. In the high technology innovation region (>8.7320), the effect of new energy development on $CO_2$ emissions is also negative and significant, but significantly smaller (-0.8141) than in the low technology innovation region (≤8.7320). Technological innovation regions (≤8.7320). In model (3), the regression results of the cross-sectional term between high technology innovation regions (>8.7320) and new energy development in the full sample DID method indicate that the negative impact of new energy development on $CO_2$ emissions is more significant in high technology innovation regions (>8.7320). Therefore, **H3** was proofed. Meanwhile, models (4)-(6) in Table 6, which are dynamic models with first-order lags of the natural logarithm of $CO_2$ emissions, are consistent with models (1)-(3) in Table 6. Then, the robustness of **H3** is verified.

**Table 6. Results of the heterogeneity analysis of new energy development to reduce $CO_2$ emissions.**

| | (1) | (2) | (3) | (4) | (5) | (6) |
|---|---|---|---|---|---|---|
| | Low technology innovation regions ($\leq$8.7320) | High technology innovation regions ($>$8.7320) | All data | Low technology innovation regions ($\leq$8.7320) | High technology innovation regions ($>$8.7320) | All data |
| First order lag term of log of $CO_2$ emissions | | | | 0.6803*** | 0.0613 | 0.4711*** |
| | | | | (14.44) | (0.75) | |
| new energy development | -0.2441* | -0.8141*** | -0.3661*** | -0.0620 | -0.7765*** | -0.2040* |
| | (-1.74) | (-4.16) | (-3.01) | (-0.63) | (-3.84) | (-1.92) |
| High technology innovation regions ($>$8.7320)×new energy development | | | -0.3410*** | | | -0.2120*** |
| | | | (-5.86) | | | (-4.11) |
| High technology innovation regions ($>$8.7320) | | | 0.1007*** | | | 0.0542*** |
| | | | (4.59) | | | (2.79) |
| Control variables | ✓ | ✓ | ✓ | ✓ | ✓ | ✓ |
| Constant | 18.9679*** | 10.5520* | 12.3533*** | 5.0679*** | 10.1294* | 6.5661*** |
| | (11.56) | (1.87) | (9.29) | (3.40) | (1.78) | (5.20) |
| Observation | 242 | 164 | 406 | 242 | 164 | 406 |
| R-squared | 0.8131 | 0.9184 | 0.9011 | 0.9101 | 0.9188 | 0.9265 |
| Province effect | Yes | Yes | Yes | Yes | Yes | Yes |
| Year effect | Yes | Yes | Yes | Yes | Yes | Yes |

Note: High technology innovation regions ($>$8.7320) indicate a dummy variable that takes the value of 1 when the log of the number of patent applications is $>$8.7320, otherwise it takes the value of 0.

## 4.6 Robust analysis

To further examine the issue of endogeneity in regression analysis, we introduce the instrumental variable approach based on the fixed effects model (IV-FE). And the natural logarithm of pm2.5 and thermal power generation, which are highly correlated with new energy development in the correlation test, are used as instrumental variables. The Sargan-Hansen overidentification test [42] and the endogeneity test are applied to check the validity of the instrumental variable approach based on the fixed effects model. The Sargan-Hansen tests for models (1) and (2) in Table 7 were non-significant, implying that these instrumental variables did not pass the over-identification test and accepting the null hypothesis that the instrumental variables are strictly exogenous, i.e. the instrumental variables are not correlated with the error terms. Therefore the natural logarithm of pm2.5 and thermal power generation are valid instrumental variables. In contrast, significant results of Davidson-MacKinnon test reject the original hypothesis (the results of instrumental variable approach based on the fixed effects model is significantly different from the original models). It means that the original models has endogeneity problems, and the instrumental variable approach is necessary. But the results of the instrumental variables approach based on the fixed effects model are consistent, which further suggests the robustness of new energy development to reduce $CO_2$ emissions. Models (3) and (4) in Table 7, multiple linear regression with additional control variables (telephone penetration, internet penetration and share of secondary industry) and a fixed effects model controlled by time trends and individual trends respectively, both show a significant negative effect of new energy development on $CO_2$ emissions, which further demonstrates the robustness of the paper's basic findings.

**Table 7. Robust test.**

| | (1) | (2) | (3) | (4) |
|---|---|---|---|---|
| | IV-FE | IV-FE | OLS | FE |
| The log of CO2 emissions | | 0.5286*** | | |
| | | (13.33) | | |
| New energy development | -1.2293*** | -0.6474*** | -0.9696*** | -0.7134*** |
| | (-8.98) | (-5.68) | (-16.04) | (-6.28) |
| The log of total electricity generation | 0.6489*** | 0.3020*** | 0.4676*** | 0.4430*** |
| | (22.86) | (8.90) | (13.80) | (12.06) |
| The log of forest area | 0.1073*** | 0.0484 | -0.1062*** | -0.0303 |
| | (2.78) | (1.57) | (-6.09) | (-0.72) |
| The log of population | -0.4158*** | -0.1903 | 0.4674*** | -0.6034*** |
| | (-2.84) | (-1.63) | (13.78) | (-4.19) |
| The log of road mileage | 0.1462*** | 0.0394* | 0.1549*** | 0.0344 |
| | (5.67) | (1.80) | (5.69) | (0.99) |
| The log of land area | 0.0315* | 0.0226 | 0.0737*** | 0.0244 |
| | (1.70) | (1.54) | (3.96) | (1.50) |
| Unemployment rate | 0.0046 | 0.0012 | 0.1299*** | 0.0096 |
| | (0.24) | (0.08) | (6.50) | (0.52) |
| Consumer price index | 0.0015 | 0.0006 | -0.0047 | -0.0252*** |
| | (0.46) | (0.24) | (-0.81) | (-3.20) |
| Growth rate of GDP | -0.0037 | 0.0035 | 0.0010 | 0.0018 |
| | (-1.33) | (1.53) | (0.18) | (0.47) |
| Telephone penetration | | | 0.0051*** | -0.0019** |
| | | | (7.67) | (-2.50) |
| Internet penetration | | | -0.0037** | -0.0029*** |
| | | | (-2.28) | (-2.65) |
| Share of secondary industry | | | 0.0051** | -0.0052*** |
| | | | (2.45) | (-2.75) |
| Constant | 6.3439*** | 3.2326*** | 0.5841 | 14.0562*** |
| | (5.63) | (3.51) | (0.88) | (9.91) |
| Observation | 406 | 406 | 378 | 378 |
| R-squared | 0.8577 | 0.9113 | 0.936 | 0.902 |
| Sargan-Hansen test | 4.31 | 1.07 | | |
| Davidson-MacKinnon test | 47.10*** | 16.29*** | | |

# 5. Conclusion, policy implications and future outlook

## 5.1 Conclusion

This paper presents an in-depth study on the reduction of $CO_2$ emissions by regional new energy development based on 18 years of statistical yearbook data from 2000 to 2017 for 30 provinces, cities and autonomous regions in China (excluding Hong Kong, Macao, Taiwan and Tibet Autonomous Region).

Firstly, the multiple regression model, the FGLS model with heteroskedasticity and the fixed effects model all confirm that new energy development has a significant negative effect on $CO_2$ emissions, i.e., new energy development significantly reduces regional $CO_2$ emissions. In the dynamic model, this effect has a significant time lag effect. It means that the new energy development cannot be quickly and effectively used to replace traditional fossil energy in the short term. So, the results show a significant positive effect in the short term, but significant

negative effect in the long term between new energy development and $CO_2$ emissions. The results further validate the unanimous conclusion of academic research that cleaner renewable energy sources are effective in reducing $CO_2$ emissions [43, 44]. This conclusion remains valid in China as well [45].

Secondly, the impact of new energy development on $CO_2$ emissions is mainly through two aspects: 1) the new energy industry is a high capital investment and high technology-intensive industry, the investment in the industry has a significant positive impact on regional economic development, and for developing countries in China, the regional economic development will to a certain extent promote $CO_2$ emissions; 2) The new energy development, as clean energy sources, reduce $CO_2$ emissions by replacing the demand for fossil energy. The introduction of a quadratic term for new energy development in this paper, with a significant negative regression coefficient, demonstrates that new energy development has an inverted U-shaped effect on $CO_2$ emissions, with new energy development at around 25% being the threshold of the inverted U-shaped effect. In other words, when new energy development is relatively low (new energy development <25%), the positive effect of $CO_2$ emissions brought about by new energy development through economic development is greater than the negative effect of $CO_2$ emissions brought about by fossil energy substitution. The overall effect of new energy development on $CO_2$ emissions is positive. When new energy development reaches a certain stage (25%), new energy development is gradually maturing and taking a leading role in economic development and energy consumption. In this case, their higher energy efficiency and substitution of highly polluting fossil energy sources of new energy development are constantly being magnified. Therefore, new energy development is gradually demonstrating $CO_2$ emission reductions, and the impact will increase as new energy development improving. A lot of researches show that the income-based environmental Kuznets curve hypothesis is not applicable to China at this stage [46, 47], but this paper empirically confirms the existence of the Environmental Kuznets Curve of $CO_2$ emissions based on new energy development.

Thirdly, in order to alleviate the impact of economic development on the continuous rise of $CO_2$ emissions, this paper puts forward the moderating effect of technological innovation on new energy development to reduce $CO_2$ emissions, accelerates technological innovation, as well as the application of technological innovation in the field of new energy, reduces the degree of new energy development and improves the efficiency of new energy development, which will be conducive to increasing the influence of new energy development on $CO_2$ emissions. Some existing studies have discussed the significant effect of technological innovations on the reduction of $CO_2$ emissions [48, 49]. This paper further analyses how technological innovations can indirectly contribute to the reduction of $CO_2$ emissions by improving the energy efficiency of new energy sources and the substitution rate of fossil energy sources.

## 5.2 Policy implications

Faced with the current situation of intensifying global warming and frequent extreme climate phenomena, reducing $CO_2$ emissions has become a common challenge faced by all countries. As the world's largest emitter of carbon dioxide, China actively promotes the development of new energy and has made important contributions to reducing carbon emissions. Based on the research conclusions of this paper, China should further increase the carbon reduction effect of new energy development from the following two aspects: 1) The government should accelerate energy transformation, further improve fiscal and financial policies to support the development of new energy, and promote solar, wind, and hydro energy Wait for the development of renewable energy and reduce dependence on fossil energy. 2) The government should make full use of financial subsidies, preferential taxation, policy guidance and other means to

increase R&D investment in the new energy industry and promote high-quality development of new energy in the new era. The upgrading and iteration of new energy technologies will greatly improve energy efficiency and thereby reduce CO$_2$ emissions.

## 5.3 Research limitations and future outlook

This paper uses China's provincial panel data to confirm the significant negative relationship between new energy development and carbon emissions, and the impact is inverted U-shaped. As for the paths through which new energy development affects carbon emissions, the authors only provide a theoretical analysis and does not empirically test the effectiveness of each path. Secondly, the authors only consider the moderating effect of technological innovation on the emission reduction effect of new energy development. The impact of other exogenous factors such as policy, society, and market need to be further explored in subsequent research.

## Author Contributions

**Conceptualization:** Feng Xiong.

**Data curation:** HuiDong Mo.

**Formal analysis:** Feng Xiong.

**Funding acquisition:** Feng Xiong.

**Investigation:** HuiDong Mo.

**Methodology:** Feng Xiong.

**Project administration:** Feng Xiong.

**Resources:** HuiDong Mo.

**Software:** HuiDong Mo.

**Supervision:** Feng Xiong.

**Visualization:** HuiDong Mo.

**Writing – original draft:** HuiDong Mo.

**Writing – review & editing:** Feng Xiong.

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
