## [Decision Letter · Decision Letter 0]

26 Sep 2023

PONE-D-23-19971How can new energy development reduce CO2 emissions: empirical evidence of inverted U-shaped relationship in ChinaPLOS ONE

Dear Dr. xiong,

Thank you for submitting your manuscript to PLOS ONE. After careful consideration, we feel that it has merit but does not fully meet PLOS ONE’s publication criteria as it currently stands. Therefore, we invite you to submit a revised version of the manuscript that addresses the points raised during the review process.

We look forward to receiving your revised manuscript.

Kind regards,

Najabat Ali, Ph.D.

Academic Editor

PLOS ONE

Journal Requirements:

3. Thank you for stating the following financial disclosure: "NO"

5. We note you have included a table to which you do not refer in the text of your manuscript. Please ensure that you refer to Table 7 in your text; if accepted, production will need this reference to link the reader to the Table.

6. We notice that your supplementary figures/tables are included in the manuscript file (embedded as the figures and tables itself). Please remove and upload your supporting informations with the file type 'Supporting Information'. Please ensure that each Supporting Information file has a legend listed in the manuscript after the references list.

**Additional Editor Comments:**

You requested to carefully revise your manuscript according to the suggestions of reviewers.

Reviewers' comments:

Reviewer's Responses to Questions

**Comments to the Author**

1. Is the manuscript technically sound, and do the data support the conclusions?

Reviewer #1: Yes

Reviewer #2: Yes

2. Has the statistical analysis been performed appropriately and rigorously? 

Reviewer #1: Yes

Reviewer #2: Yes

3. Have the authors made all data underlying the findings in their manuscript fully available?

Reviewer #1: Yes

Reviewer #2: Yes

4. Is the manuscript presented in an intelligible fashion and written in standard English?

Reviewer #1: Yes

Reviewer #2: Yes

5. Review Comments to the Author

Reviewer #1: It is my pleasure to review the current manuscript having title “How can new energy development reduce CO2 emissions: empirical evidence of inverted U-shaped relationship in China” for the esteemed journal. The manuscript has some merits but there are certain issues which need to be addressed to improve the quality of the manuscript. The authors must consider the following suggestions:

1. The abstract is well-structured. However, it should further underscore the scientific value added of your paper in your abstract.

2. The novelty of this paper should be further justified by highlighting main contributions of the paper.

3. More recent and relevant papers should be cited to enrich the literature.

4. In the methodology section, the model of the study needs to be justified by comparing it with other models.

5. In the results section, the obtained results should be compared with existing studies in the field.

6. Grammar check is required to avoid any possible English errors.

Reviewer #2: Here are the comments relevant to paper;

Abstract: abstract is very well written and clearly explain all the things.

Introduction: The novelty of this paper should be justified by highlighting the main contributions of the article.

Literature review: Literature review is well written, However, more recent and related papers should be cited such as;

https://doi.org/10.1016/j.techsoc.2023.102364

Methodology: This section is very well written and explained

Conclusion: This section has required more attention: Conclusion ,policy implication and limitations should be written in separate paragraph .

A grammar check is also required

6. PLOS authors have the option to publish the peer review history of their article (what does this mean?). If published, this will include your full peer review and any attached files.

Reviewer #1: **Yes: **Shahbaz Tariq

Reviewer #2: No

---

## [Author Response · Author response to Decision Letter 0]

9 Nov 2023

Dear Editor and dear reviewers, 

Thank you for your letter and the reviewers’ comments concerning our manuscript entitled "How can new energy development reduce CO2 emissions: empirical evidence of inverted U-shaped relationship in China" (PONE-D-23-19971). Those comments are valuable and very helpful. We have read through comments carefully and have made corrections. Based on the instructions provided in your letter, we uploaded the file of the revised manuscript, which named "Revised Manuscript with Track Changes". Revisions in the text are shown using red for additions, and strikethrough font for deletions. The responses to the reviewer’s comments are presented following.

We would love to thank you for allowing us to resubmit a revised copy of the manuscript and we highly appreciate your time and consideration.

Data Availability statement

The data set of this paper are upload in Harvard Dataverse, and the DOIs is https://doi.org/10.7910/DVN/YDX6X9.

Sincerely,

Feng Xiong

Reviewer #1

Q1. The abstract is well-structured. However, it should further underscore the scientific value added of your paper in your abstract.

Response: Many thanks for your comments. We add further elaboration of the scientific value of the conclusions in the abstract. In response to the conclusion of “inverted U-shaped relationship between new energy development and CO2 emissions”, the paper adds the statement “which confirms the existence of the Environmental Kuznets Curve (EKC) of CO2 emissions based on new energy development”. And in response to the conclusion of “technological innovation has a significant moderating effect on the CO2 emission reduction effect of new energy development”, the paper adds the statement “which confirms theoretically the importance of technological innovation in accelerating new energy substitution and improving energy efficiency”.

Q2. The novelty of this paper should be further justified by highlighting main contributions of the paper.

Response: Many thanks for your comments. This paper further adds to the main contributions of the paper on page 4, lines 7 to 13. The content is written as, “In summary, this paper adds to the existing researches on the relationship between new energy development and carbon emissions from the following three main aspects: 1) The new energy development in China has already realized a CO2 emissions reduction effect. 2) The impact of new energy development on carbon emissions is non-linear, i.e. Environmental Kuznets Curve (EKC) of CO2 emissions based on new energy development exists. 3) Technological innovation can contribute to the CO2 emissions reduction effect of the new energy development.”

Q3. More recent and relevant papers should be cited to enrich the literature.

Response: Many thanks for your comments. The paper is further supplemented with relevant recent and cutting-edge researches in the "literature review" section. The cited articles are as follows:

Xu, D.; Abbas, S.; Rafique, K.; Ali, N. The Race to Net-Zero Emissions: Can Green Technological Innovation and Environmental Regulation Be the Potential Pathway to Net-Zero Emissions? Technology in Society 2023, 75, 102364, doi:10.1016/j.techsoc.2023.102364.

2. Lin, B.; Zhang, Q. Green Technology Innovation under Differentiated Carbon Constraints: The Substitution Effect of Industrial Relocation. Journal of Environmental Management 2023, 345, 118764, doi:10.1016/j.jenvman.2023.118764.

3. Liu, D.; Zhu, X.; Wang, Y. China’s Agricultural Green Total Factor Productivity Based on Carbon Emission: An Analysis of Evolution Trend and Influencing Factors. Journal of Cleaner Production 2021, 278, 123692, doi:10.1016/j.jclepro.2020.123692.

4. He, L.; Zhang, X.; Yan, Y. Heterogeneity of the Environmental Kuznets Curve across Chinese Cities: How to Dance with ‘Shackles’? Ecological Indicators 2021, 130, 108128, doi:10.1016/j.ecolind.2021.108128.

5. Sun, Y.; Li, M.; Zhang, M.; Khan, H.S.U.D.; Li, J.; Li, Z.; Sun, H.; Zhu, Y.; Anaba, O.A. A Study on China’s Economic Growth, Green Energy Technology, and Carbon Emissions Based on the Kuznets Curve (EKC). Environ Sci Pollut Res 2021, 28, 7200–7211, doi:10.1007/s11356-020-11019-0.

6. Iris, Ç.; Lam, J.S.L. A Review of Energy Efficiency in Ports: Operational Strategies, Technologies and Energy Management Systems. Renewable and Sustainable Energy Reviews 2019, 112, 170–182, doi:10.1016/j.rser.2019.04.069.

7. Zou, C.; Xiong, B.; Xue, H.; Zheng, D.; Ge, Z.; Wang, Y.; Jiang, L.; Pan, S.; Wu, S. The Role of New Energy in Carbon Neutral. Petroleum Exploration and Development 2021, 48, 480–491, doi:10.1016/S1876-3804(21)60039-3.

Q4. In the methodology section, the model of the study needs to be justified by comparing it with other models.

Response: Many thanks for your comments. Among the empirical models, the classical multiple regression model and the fixed effects model have been elaborated in this paper. Meanwhile, in the process of empirical analysis, the regression results of the two models have also been compared and the empirical results are consistent. In order to further test the robustness of the regression results, this paper also uses the instrumental variable approach based on the fixed effect model (IV-FE) to test the endogeneity of the model in the "Robust analysis" section, and the results still show that the conclusions of this paper are consistent.

Q5. In the results section, the obtained results should be compared with existing studies in the field.

Response: Many thanks for your comments. In the conclusions section, the paper highlights the contribution of the findings of this paper by comparing each of the conclusions with existing research. Additional comparisons of the conclusions are on page 18, lines 16 to 21, page 19, lines 15 to 18, and lines 25 to 30.

Q6. Grammar check is required to avoid any possible English errors.

Response: Many thanks for your comments. We have made detailed revisions to the grammar of this paper.

Reviewer #2

Q1. Abstract: abstract is very well written and clearly explain all the things.

Response: Thank you for your high approval of this content!

Q2. Introduction: The novelty of this paper should be justified by highlighting the main contributions of the article.

Response: Many thanks for your comments. This paper further adds to the main contributions of the paper on page 4, lines 7 to 13. The content is written as, “In summary, this paper adds to the existing researches on the relationship between new energy development and carbon emissions from the following three main aspects: 1) The new energy development in China has already realized a CO2 emission reduction effect. 2) The impact of new energy development on carbon emissions is non-linear, i.e. Environmental Kuznets Curve (EKC) of CO2 emissions based on new energy development exists. 3) Technological innovation can contribute to the CO2 emissions reduction effect of the new energy development.”

Q3. Literature review: Literature review is well written, However, more recent and related papers should be cited such as https://doi.org/10.1016/j.techsoc.2023.102364.

Response: Many thanks for your comments. The paper is further supplemented with relevant recent and cutting-edge researches in the "literature review" section. The cited articles are as follows:

1. Xu, D.; Abbas, S.; Rafique, K.; Ali, N. The Race to Net-Zero Emissions: Can Green Technological Innovation and Environmental Regulation Be the Potential Pathway to Net-Zero Emissions? Technology in Society 2023, 75, 102364, doi:10.1016/j.techsoc.2023.102364.

2. Lin, B.; Zhang, Q. Green Technology Innovation under Differentiated Carbon Constraints: The Substitution Effect of Industrial Relocation. Journal of Environmental Management 2023, 345, 118764, doi:10.1016/j.jenvman.2023.118764.

3. Liu, D.; Zhu, X.; Wang, Y. China’s Agricultural Green Total Factor Productivity Based on Carbon Emission: An Analysis of Evolution Trend and Influencing Factors. Journal of Cleaner Production 2021, 278, 123692, doi:10.1016/j.jclepro.2020.123692.

4. He, L.; Zhang, X.; Yan, Y. Heterogeneity of the Environmental Kuznets Curve across Chinese Cities: How to Dance with ‘Shackles’? Ecological Indicators 2021, 130, 108128, doi:10.1016/j.ecolind.2021.108128.

5. Sun, Y.; Li, M.; Zhang, M.; Khan, H.S.U.D.; Li, J.; Li, Z.; Sun, H.; Zhu, Y.; Anaba, O.A. A Study on China’s Economic Growth, Green Energy Technology, and Carbon Emissions Based on the Kuznets Curve (EKC). Environ Sci Pollut Res 2021, 28, 7200–7211, doi:10.1007/s11356-020-11019-0.

6. Iris, Ç.; Lam, J.S.L. A Review of Energy Efficiency in Ports: Operational Strategies, Technologies and Energy Management Systems. Renewable and Sustainable Energy Reviews 2019, 112, 170–182, doi:10.1016/j.rser.2019.04.069.

7. Zou, C.; Xiong, B.; Xue, H.; Zheng, D.; Ge, Z.; Wang, Y.; Jiang, L.; Pan, S.; Wu, S. The Role of New Energy in Carbon Neutral. Petroleum Exploration and Development 2021, 48, 480–491, doi:10.1016/S1876-3804(21)60039-3.

Q4. Methodology: This section is very well written and explained

Response: Thank you for your high approval of this section!

Q5. Conclusion: This section has required more attention: Conclusion ,policy implication and limitations should be written in separate paragraph.

Response: Many thanks for your comments. The policy implications are written in separate paragraph on page 20, lines 1 to 15. And the limitations of the paper are written on page 20, lines 16 to 24.

Q6. A grammar check is also required.

Response: Many thanks for your comments. We have made detailed revisions to the grammar of this paper.

---

## [Editor Report · Decision Letter 1]

13 Nov 2023

How can new energy development reduce CO2 emissions: empirical evidence of inverted U-shaped relationship in China

PONE-D-23-19971R1

Dear Dr. Xiong,

We’re pleased to inform you that your manuscript has been judged scientifically suitable for publication and will be formally accepted for publication once it meets all outstanding technical requirements.

Kind regards,

Najabat Ali, Ph.D.

Academic Editor

PLOS ONE
---

## [Editor Report · Acceptance letter]

15 Nov 2023

PONE-D-23-19971R1 

How can new energy development reduce CO_2_ emissions: empirical evidence of inverted U-shaped relationship in China 

Dear Dr. Xiong:

I'm pleased to inform you that your manuscript has been deemed suitable for publication in PLOS ONE. Congratulations! Your manuscript is now with our production department. 

Kind regards, 

on behalf of

Dr. Najabat Ali 

Academic Editor

PLOS ONE